# Advanced imaging and trends in hospitalizations from the emergency department

Shih-Chuan Chou[1]*, Justine M. Nagurney[2], Jeremiah D. Schuur[3], Scott G. Weiner[1]

1 Department of Emergency Medicine, Brigham and Women's Hospital, Boston, Massachusetts, United States of America, 2 Department of Emergency Medicine, Beth Israel Deaconess Medical Center, Boston, Massachusetts, United States of America, 3 Department of Emergency Medicine, Brown Alpert Medical School, Providence, Rhode Island, United States of America

* schou2@bwh.harvard.edu

## Abstract

**Data Availability Statement:** The data and accompanying documentation used for this paper is publicly accessible through the NCHS website at https://www.cdc.gov/nchs/ahcd/datasets_documentation_related.htm.

### Objective

The proportion of US emergency department (ED) visits that lead to hospitalization has declined over time. The degree to which advanced imaging use contributed to this trend is unknown. Our objective was to examine the association between advanced imaging use during ED visits and changes in ED hospitalization rates between 2007–2008 and 2015–2016.

### Methods

We analyzed data from the National Hospital Ambulatory Medical Care Survey. The primary outcome was ED hospitalization, including admission to inpatient and observation units and outside transfers. The primary exposure was advanced imaging during the ED visit, including computed tomography, magnetic resonance imaging, and ultrasound. We constructed a survey-weighted multivariable logistic regression with binary outcome of ED hospitalization to examine changes in adjusted hospitalization rates from 2007–2008 to 2015–2016, comparing ED visits with and without advanced imaging.

### Results

ED patients who received advanced imaging (versus those who did not) were more likely to be 65 years or older (25.3% vs 13.0%), non-Hispanic white (65.3% vs 58.5%), female (58.4% vs 54.1%), and have Medicare (26.5% vs 16.0%). Among ED visits with advanced imaging, adjusted annual hospitalization rate declined from 22.5% in 2007–2008 to 17.3% (adjusted risk ratio [aRR] 0.77; 95% CI 0.68, 0.86) in 2015–2016. In the same periods, among ED visits without advanced imaging, adjusted annual hospitalization rate declined from 14.3% to 11.6% (aRR 0.81; 95% CI 0.73, 0.90). The aRRs between ED visits with and without advanced imaging were not significantly different.

**Funding:** SCC and JDS received support from the Emergency Medicine Foundation's ED clinical work intensity grant. The sponsors had no role in the study design, data collection and analysis, decision to publish, or preparation of the manuscript. https://www.emfoundation.org/.

**Competing interests:** The authors have declared that no competing interests exist.

## Conclusion

From 2007–2016, ED visits with advanced imaging did not have a greater reduction in admission rate compared to those without advanced imaging. Our results suggest that increasing advanced imaging use likely had a limited role in the general decline in hospital admissions from EDs. Future research is needed to further validate this finding.

## Introduction

### Background

Advanced imaging has become an integral part of the modern emergency department (ED). By 2005 EDs across the US have had near universal access to computed tomography (CT) with a growing presence of magnetic resonance imaging (MRI) availability [1]. The use of advanced imaging grew exponentially over the early 2000s, with imaging rates tripling between 1996 and 2007 [2]. The increased use of advanced imaging have raised concerns about the negative impacts of overuse, including radiation exposure associated with CT scans [3] and their associated high costs. However, whether the increased use of advanced imaging has provided additional value remains debated.

One way to evaluate the value of increased advanced imaging utilization in the ED is whether it has led to cost savings through avoided hospitalizations. With inpatient care accounting for nearly one-third of the US national health expenditure [4], numerous policies have aimed to reduce inpatient hospitalizations, including the 2010 Recovery Audit Contractor program and 2013 Two-Midnight Rule disincentivized short-stay admissions [5]. These policies have created a substantial pressure on EDs to shift care towards the outpatient setting as hospitalizations originating from the ED increased to more than 80% of hospitalizations by 2009 [6]. From 2006 to 2014, while ED visits increased by 18%, hospitalization rates of ED visits have declined by nearly 10% [7]. Although this decrease in hospitalization rates coincided with the rapid expansion of advanced imaging use, whether advanced imaging has contributed to the declining admission rates remains unexplored.

In this study, we utilized the data from a nationally representative sample of US ED visits to examine the association between advanced imaging use and the trends in ED hospitalization rates. Prior studies have shown that advanced imaging use rose sharply between 1997 to 2007 [2], but, to our knowledge, no study have examined whether this growth has continued. Furthermore, the potential link between advanced imaging and the decrease in ED hospitalization rates has not been examined. We hypothesized that, compared to ED visits without advanced imaging, ED visits with advanced imaging were associated with a greater decline in admission rate.

## Methods

### Dataset

We analyzed the cross-sectional data of the National Hospital Ambulatory Medical Care Survey ED sample (NHAMCS-ED), a multistage, probability sample of US ED visits administered by the National Center for Health Statistics (NCHS) from 2007 to 2016. NHAMCS-ED uses a four-stage sampling design: 1) county-level geographic region as primary sampling units (PSU), 2) hospitals within each PSU, 3) emergency service areas served by each hospital, and 4) 100–150 patient records from a randomly assigned four-week period of the survey year

within each emergency service area. NCHS excluded federal, military, and Veterans Administration hospitals. Final samples included from 267 to 408 responding EDs reporting a total of 25,000 to 30,000 ED visits annually. Probability weights and survey design variables were assigned to every visit to allow the calculation of nationally representative estimates and standard errors. Full details of the NHAMCS methodology are available online [8]. This study was exempt from review by the institutional review board of the authors' institutions.

## Outcomes

The primary outcome of interest is ED hospitalization. We defined ED hospitalization as inpatient admission, observation stay, or hospital transfer. We categorized observation stays as hospitalization because it reflects the ED physician's determination that patients could not be safely discharged. Furthermore, in absence of an ED-based observation unit, patients would often be cared for in an hospital floor setting indistinguishable from inpatient care [9]. We also considered hospital transfer as equivalent to the decision to hospitalize. In transferring, the ED clinicians likely believed that there was a need for higher levels of care and the patients could not be safely discharged.

## Key variables

We defined advanced imaging to include CT, MRI, and ultrasound. Owing to a lack of direct potential harm and relatively lower costs, ultrasound is often omitted when examining the use and overuse of advanced imaging. However, in this context, we included ultrasound because, like CT and MRI, it is an imaging modality that is high-cost and often not immediately available in the outpatient context.

NHAMCS data contains patients' presenting symptoms or complaints. Previous studies examining the value of care have often been limited by retrospective administrative claims data which contains only the diagnoses obtained after a completed medical evaluation. This limitation is highlighted by National Quality Forum's recent move towards complaint-based quality measures [10]. We adopted the definitions developed by Kocher et al, who identified the 20 most common presenting symptoms using the *primary reasons for visit* variable in NHAMCS (S1 Table) [2].

We included patient characteristics as covariates in our analysis, including patient age, sex, race/ethnicity, and insurance status. We combined the indicators for self-reported race and ethnicity to generate race categories of non-Hispanic white, non-Hispanic Black, Hispanic, and others. We defined patient insurance status using the multinomial variable "expected payment type." In most years, NCHS used a hierarchy that assigned visits by Medicaid and Medicare dual-eligible beneficiaries to Medicare. But in data year 2007, this hierarchy was different in that these visits were assigned to Medicaid. To maintain consistency, we reassigned patients visits in 2007 with dual Medicare and Medicaid coverage to Medicare.

We also used visit characteristics as covariates, including whether the visit was seen by a physician assistant or nurse practitioner (PA/NP), whether a resident was among the physician team, and hospital geographic region. Triage category has changed in NHAMCS over time. To minimize inconsistency, we collapsed the categories into urgent/emergent and others. We identified the arrival time and day of the week for each visit and categorized each visit as weekday, defined as 8AM to 5PM, Monday through Friday, and nights/weekends, defined as all other hours outside of weekdays.

## Statistical analysis

All analyses incorporated survey design and weights assigned within NHAMCS. We first calculated the weighted proportion of ED visits across patient and visit characteristics, stratified by whether advanced imaging was obtained during the ED visit. We then calculated and plotted imaging and hospitalization rates in bi-annual intervals.

We modeled the probability of receiving advanced imaging during ED visits as a binary outcome using survey-weighted multivariable logistic regression, with an indicator for 2007–2008 versus 2015–2016, controlling for patient and visit characteristics. We used the marginal estimating method to calculate the probability of receiving advanced imaging in the 2007–2008 and 2015–2016 time periods and then calculated the adjusted risk ratio.

Next, we used a survey-weighted multivariable logistic regression to model the probability of hospital admission versus discharge. To examine the association between advanced imaging and the trends in hospitalization, we included an indicator for receiving advanced imaging, an indicator for 2007–2008 versus 2015–2016, and the interaction between the two, controlling for patient and visit characteristics. We used marginal estimating method to calculate adjusted annual hospitalization rates for visits with and without advanced imaging in 2007–2008 and 2015–2016, as well as the adjusted risk ratio and relative proportional change in hospitalization rates comparing visits with and without advanced imaging.

We repeated the analysis for visits with each of the 20 most common presenting symptoms. We also performed sensitivity checks with two different specification, 1) only accounting for inpatient admissions and 2) only considering CT/MRI as advanced imaging. All tests were two-sided, and we considered an alpha of less than 0.01 as significant, consistent with NCHS-recommended practices. STATA 15/MP (College Station, TX) was used for all analyses.

## Results

### Study population

Between 2007 and 2016, a total of 289,188 ED visits were included in the NHAMCS dataset with 110,152 visits in the years 2007–2008 and 2015–2016. From 2007 to 2016, total ED visits in the US increased from an estimated 116.8 million annually to 145.6 million. Overall 18.9% of ED visits (95% CI 18.4–19.5) included advanced imaging. Compared to ED patients who did not receive advanced imaging (Table 1), ED patients who received advanced imaging were more likely to be 45 years or older (52.3% vs 33.7%, p<0.001), female (58.4% vs 54.1, p<0.001), non-Hispanic white (65.3% vs 58.5%, p<0.001), and insured by private insurance (35.1% vs 32.6%, p<0.001) or Medicare (26.5% vs 16.0%, p<0.001). Imaged visits were slightly more likely to be during office hours (37.9% vs 34.6, p<0.001) but much more likely to be urgent or emergent (61.5% vs 39.7%, p<0.001).

### Trends in advanced imaging use

Overall advanced imaging use in the ED increased, from 17.1% (95% CI 16.2–18.0) in 2007–2008 to 21.3% (95% CI 20.2–22.4) in 2015–2016. This increase was driven by the continued growth in CT/MRI use and, to a smaller degree, by growing use of ultrasound (Fig 1).

In our modeling, we found that the overall adjusted imaging rate increased by 32% (adjusted risk ratio [aRR] 1.32, 95% CI 1.23–1.40, p<0.001; Table 2). Among the 20 most common presenting complaints, the adjusted advanced imaging rate increased significantly in ED visits for injury (aRR 1.35; 95% CI 1.23–1.48, p<0.001), upper respiratory symptoms (aRR 1.54; 95% CI 1.13–1.95, p = 0.009), abdominal pain (aRR 1.18; 95% CI 1.08–1.28, p<0.001), leg symptoms (aRR 1.35; 95% CI 1.13–1.57, p = 0.002), neck/back pain (aRR 1.39; 95% CI 1.18–

**Table 1. Demographic characteristics of study population by use of advanced imaging, 2007 to 2016.**

| Characteristics | Advanced Imaging (n = 52,942) | | | No Advanced Imaging (n = 236,246) | | | |
| --- | --- | --- | --- | --- | --- | --- | --- |
| | n | Weighted % | (95% CI) | n | Weighted % | (95% CI) | P |
| Age | | | | | | | <0.001 |
| <15 | 2,797 | 5.4 | (5.0, 5.8) | 51,274 | 22.3 | (21.3, 23.3) | |
| 15–24 | 7,162 | 13.5 | (12.9, 14.1) | 37,408 | 15.9 | (15.6, 16.2) | |
| 25–44 | 15,509 | 28.9 | (28.3, 29.5) | 67,183 | 28.0 | (27.5, 28.5) | |
| 45–64 | 14,211 | 27.0 | (26.4, 27.7) | 49,647 | 20.7 | (20.3, 21.1) | |
| 65–74 | 5,128 | 10.0 | (9.6, 10.4) | 13,556 | 5.8 | (5.6, 6.1) | |
| > = 75 | 8,135 | 15.3 | (14.7, 15.9) | 17,178 | 7.2 | (6.9, 7.6) | |
| Female | 31,005 | 58.4 | (57.7, 59.1) | 126,561 | 54.1 | (53.8, 54.5) | <0.001 |
| Race | | | | | | | <0.001 |
| Non-Hispanic White | 33,837 | 65.3 | (63.6, 67.0) | 135,410 | 58.5 | (56.5, 60.4) | |
| Non-Hispanic Black | 9,602 | 18.2 | (16.6, 20.0) | 55,344 | 23.2 | (21.1, 25.3) | |
| Hispanic | 7,498 | 13.6 | (12.4, 14.9) | 36,676 | 15.4 | (14.0, 16.8) | |
| Other | 2,005 | 2.8 | (2.4, 3.3) | 8,816 | 3.1 | (2.7, 3.5) | |
| Insurance | | | | | | | <0.001 |
| Private[a] | 18,581 | 35.1 | (34.0, 36.2) | 77,648 | 32.6 | (31.5, 33.6) | |
| Medicare | 14,033 | 26.5 | (25.6, 27.4) | 37,662 | 16.0 | (15.4, 16.6) | |
| Medicaid | 10,687 | 19.5 | (18.6, 20.5) | 70,463 | 29.2 | (28.1, 30.4) | |
| Uninsured/self-pay | 6,464 | 12.0 | (11.4, 12.7) | 33,373 | 14.3 | (13.6, 15.0) | |
| Unknown | 3,177 | 6.8 | (5.7, 8.2) | 17,100 | 8.0 | (6.8, 9.3) | |
| **Visit Characteristics** | | | | | | | |
| Time of visit | | | | | | | <0.001 |
| Office Hours | 18,257 | 37.9 | (37.4–38.5) | 77,403 | 34.6 | (34.3–35.0) | |
| Weeknights | 13,655 | 27.9 | (27.3–28.4) | 65,829 | 30.1 | (29.8–30.4) | |
| Weekends | 16,836 | 34.2 | (33.7–34.7) | 77,741 | 35.3 | (35.0–35.5) | |
| Triage Level | | | | | | | <0.001 |
| Urgent/Emergent | 33,453 | 61.5 | (58.8, 64.1) | 96,443 | 39.7 | (38.2, 41.4) | |
| Non-urgent | 8,033 | 14.8 | (13.7, 16.0) | 80,607 | 34.6 | (33.2, 36.1) | |
| Unknown/Not triaged | 11,456 | 23.7 | (21.1, 26.5) | 59,196 | 25.6 | (23.2, 28.2) | |
| Seen by PA/NP | 8,000 | 15.9 | (14.4, 17.4) | 40,325 | 18.8 | (17.3, 20.3) | <0.001 |
| Seen by Resident | 6,394 | 10.5 | (9.2, 11.9) | 23,514 | 8.4 | (7.3, 9.5) | <0.001 |
| Hospital Region | | | | | | | 0.044 |
| Northeast | 11,568 | 17.8 | (15.8, 20.1) | 52,987 | 17.7 | (15.6, 20.0) | |
| Midwest | 12,608 | 23.6 | (20.4, 27.2) | 53,297 | 22.6 | (19.7, 25.8) | |
| South | 17,834 | 36.8 | (33.0, 40.9) | 83,032 | 39.2 | (35.3, 43.2) | |
| West | 10,932 | 21.8 | (19.2, 24.6) | 46,930 | 20.5 | (18.2, 23.1) | |

Private insurance status includes worker's compensation. Weeknights were defined as Mon-Thursday after 5 through 8am the next day. Weekends defined as Friday after 5pm to Monday 8am.

Abbreviations: CI, confidence interval; PA, physician assistant; NP, nurse practitioner.

1.61, p<0.001), and dizziness/syncope (aRR 1.23; 95% CI 1.06–1.40, p = 0.009). There were no presenting complaints where advanced imaging use decreased.

## Advanced imaging use and trends in hospitalization rates

Hospitalization rates declined during the study period from 16.2% (95% CI 15.0–17.4) in 2007–2008 to 12.2% (95% CI 10.9–13.7) in 2015–2016 (Fig 1). Adjusted hospitalization rates

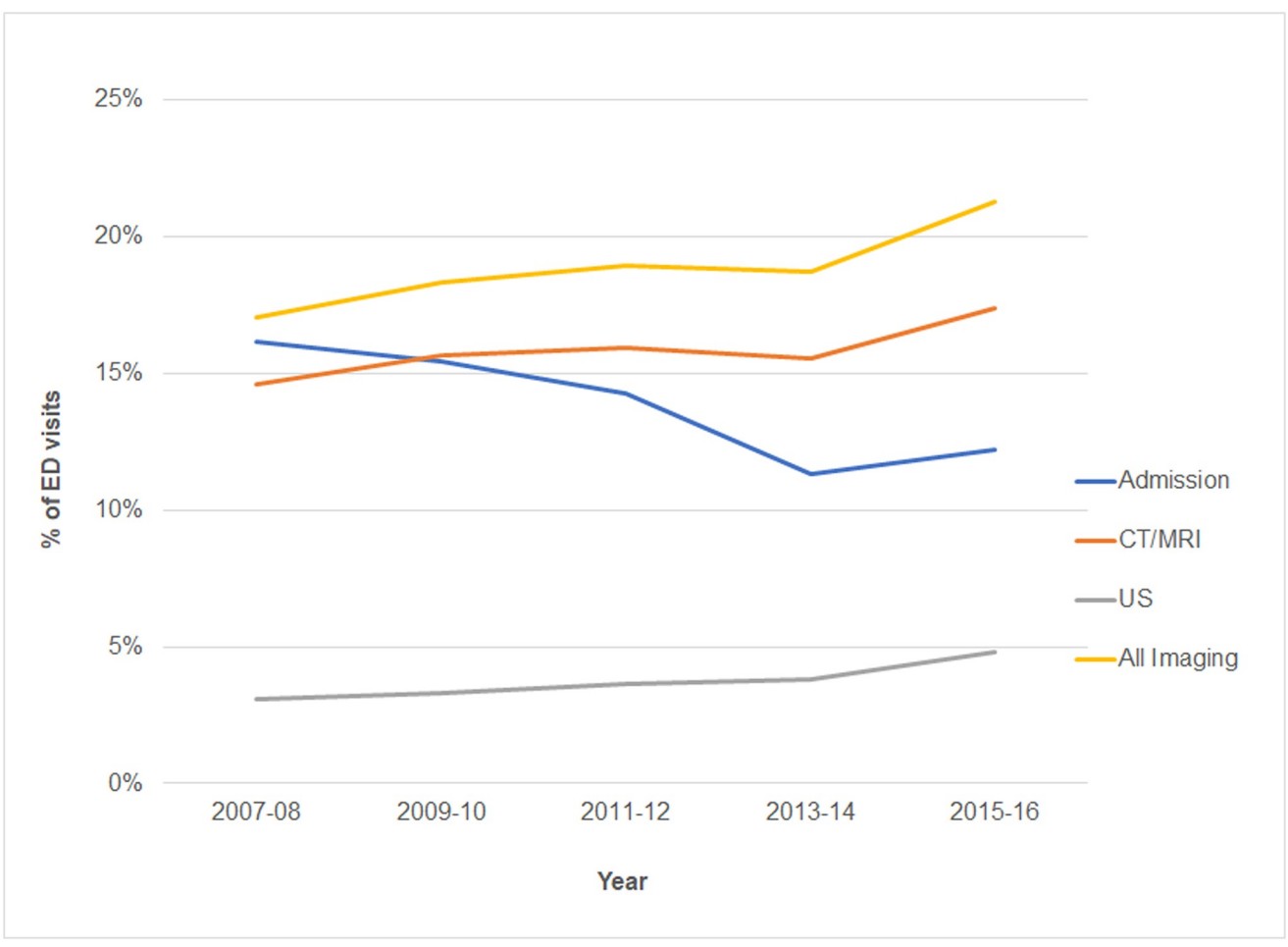

**Fig 1. Proportion of ED visits that received CT/MRI or ultrasound increase while those leading to hospital admission decreased.** Weighted bi-annual proportions calculated from the National Hospital Ambulatory Medical Care Survey.

among ED visits with and without advanced imaging decreased overall and for most presenting complaints (Table 3). Comparing hospitalization rates between ED visits with and without advanced imaging, the relative change in hospitalization rates between 2007–2008 and 2015–2016 was not significantly different (relative difference: -4.0%; 95% CI -11.2, 3.2; p = 0.27; Table 3). In the complaint-specific analyses, though no relative difference reached the *a priori* level of statistical significance at p<0.01, there were relative increases among visits for Neck/Back pain (65.3%; 95% CI 9.5, 121.2; p = 0.022), shortness of breath (30.4%; 95% CI 4.9, 55.9; p = 0.019), syncope/dizziness (29.2%; 95% CI 3.6, 54.8; p = 0.025), and general weakness (29.9%; 95% CI 3.5, 56.2; p = 0.026) that reached p<0.05. Notably, among these presenting complaints, there was a decrease in adjusted hospitalization rate among the unimaged ED visits while the adjusted hospitalization rates among visits with advanced imaging did not significantly change (Table 3).

When we only considered inpatient admission and outside transfer as hospitalizations (excluding observation admissions), the results did not materially differ (S2 Table). When we restricted the definition of advanced imaging to only CT and MRI, results also remained stable (S3 Table). Furthermore, complaint-specific analysis showed similar relative increase for Neck/Back pain, shortness of breath, and general weakness that reached p<0.05.

**Table 2. Imaging rate by Presenting symptom in 2007–2008 and 2015–2016.**

| | % of total ED visits | | | Adjusted Imaging Rate (%; 95% CI) | | | | Adjusted Risk Ratio | | |
| | | | | 2007–2008 | | 2015–2016 | | | | |
| | n | Weighted % | (95% CI) | % | (95% CI) | % | (95% CI) | | (95% CI) | p-value |
|---|---|---|---|---|---|---|---|---|---|---|
| Overall | 110,152 | | | 16.5 | (15.8, 17.3) | 21.8 | (20.7, 22.9) | **1.32** | **(1.23, 1.40)** | **<0.001** |
| Injury | 16,157 | 14.6 | (14.0, 15.2) | 14.7 | (13.7, 15.6) | 20.0 | (18.5, 21.3) | **1.35** | **(1.23, 1.48)** | **<0.001** |
| Psychiatric | 9,740 | 8.5 | (8.3, 8.8) | 32.6 | (30.4, 34.8) | 33.7 | (31.3, 36.0) | 1.03 | (0.94, 1.13) | 0.48 |
| Upper Respiratory | 9,425 | 8.7 | (8.4 9.2) | 3.0 | (2.43, 3.54) | 4.6 | (3.7, 5.5) | **1.54** | **(1.13, 1.95)** | **0.009** |
| Abdominal Pain | 8,200 | 8.0 | (7.7, 8.3) | 41.3 | (38.6, 43.9) | 48.8 | (46.3, 51.3) | **1.18** | **(1.08, 1.28)** | **<0.001** |
| Leg Symptoms | 5,600 | 5.0 | (4.8, 5.2) | 12.1 | (10.6, 13.5) | 16.3 | (14.4, 18.2) | **1.35** | **(1.13, 1.57)** | **0.002** |
| Chest pain | 5,398 | 5.3 | (5.0, 5.5) | 15.9 | (14.1, 17.7) | 17.8 | (15.4, 20.2) | 1.12 | (0.92, 1.32) | 0.25 |
| Neck/Back pain | 5,313 | 4.9 | (4.7, 5.1) | 17.1 | (15.4, 18.8) | 23.9 | (21.0, 26.7) | **1.39** | **(1.18, 1.61)** | **<0.001** |
| Fever | 4,436 | 4.2 | (3.9, 4.6) | 4.5 | (3.4, 5.6) | 5.6 | (4.2, 7.0) | 1.24 | (0.82, 1.67) | 0.26 |
| Nausea/Vomiting/Diarrhea | 4,333 | 4.1 | (3.9, 4.3) | 15.6 | (13.8, 17.4) | 19.5 | (17.2, 21.8) | 1.25 | (1.04, 1.46) | 0.018 |
| Shortness of Breath | 4,057 | 3.7 | (3.5, 3.9) | 11.9 | (10.1, 13.7) | 17.1 | (14.1, 20.1) | 1.44 | (1.10, 1.78) | 0.012 |
| Arm Symptoms | 3,854 | 3.4 | (3.2, 3.7) | 6.8 | (5.2, 8.3) | 10.3 | (8.0, 12.5) | 1.52 | (1.03, 2.00) | 0.037 |
| Headache | 3,456 | 3.3 | (3.1, 3.4) | 36.6 | (33.1, 40.0) | 38.0 | (34.6, 41.3) | 1.04 | (0.91, 1.17) | 0.57 |
| Skin Complaints | 2,961 | 2.6 | (2.5, 2.8) | 2.8 | (1.9, 3.6) | 3.8 | (2.8, 5.3) | 1.38 | (0.68, 2.09) | 0.29 |
| Dizziness/syncope | 2,284 | 2.1 | (2.0, 2.3) | 36.1 | (32.6, 39.7) | 44.4 | (40.1, 48.7) | **1.23** | **(1.06, 1.40)** | **0.009** |
| Pregnancy Problems | 1,563 | 1.1 | (1.0, 1.2) | 36.3 | (31.1, 41.5) | 46.6 | (38.5, 54.7) | 1.28 | (1.01, 1.56) | 0.046 |
| Flank Pain | 1,374 | 1.4 | (1.3, 1.4) | 48.6 | (43.3, 53.9) | 57.9 | (53.3, 62.5) | 1.19 | (1.02, 1.36) | 0.028 |
| General Weakness | 1,203 | 1.2 | (1.1, 1.3) | 29.6 | (25.8, 33.4) | 33.9 | (28.2, 39.6) | 1.14 | (0.90, 1.39) | 0.24 |
| Neurological Symptom | 1,169 | 1.0 | (1.0, 1.1) | 50.2 | (45.2, 55.1) | 55.2 | (48.8, 61.6) | 1.10 | (0.93, 1.27) | 0.25 |
| Convulsions | 1,054 | 1.0 | (0.9, 1.0) | 37.9 | (33.0, 42.9) | 39.3 | (33.5, 45.1) | 1.04 | (0.84, 1.24) | 0.71 |
| Vaginal Bleeding | 770 | 0.7 | (0.6, 0.7) | 33.2 | (27.3, 39.0) | 37.4 | (29.6, 45.2) | 1.13 | (0.81, 1.45) | 0.44 |

Adjusted imaging rate and risk ratios calculated using multivariable survey-weighted logistic regression and marginal estimation methods, adjusting for patient and visit characteristics.

Abbreviation: ED, emergency department; CI, confidence interval.

## Discussion

As EDs play an increasingly central role in the care of acute episodic illnesses, to further define the value of emergency care, we need to examine the relationship between the increased resource use in EDs and the changes in downstream costs such as reduced hospital admissions. Prior studies have found that the expansion of ED capabilities, including the rising use of advanced imaging, has occurred while ED hospitalizations declined [2, 7]. In this analysis, we found that high-cost advanced imaging use has continued to grow modestly, but advanced imaging was overall not associated with larger decline in ED hospitalization rates.

In our complaint-specific analyses, however, we found that, among ED visits for neck/back pain, syncope/dizziness, and generalized weakness, advanced imaging had weak association with higher admission rates. These findings were driven by a decrease in the hospitalization rates among visits without advanced imaging from 2007–2008 to 2015–2016, but no significant change in the hospitalization rates among visits with advanced imaging in the same time period.

Several explanations may be possible. Increased advanced imaging may have improved diagnostic yield, revealing diagnoses that may not have otherwise been detected and required hospital admission. However, improved diagnostic yield is unlikely given these associations were seen among presenting complaints where testing has been shown to have low diagnostic

**Table 3. Changes in adjusted admission rates comparing 2007–2008 to 2015–2016.**

| Presenting complaint | Visits with Imaging | | | | Visits without Imaging | | | | Relative Difference, % (95% CI) | | |
|---|---|---|---|---|---|---|---|---|---|---|---|
| | Adjusted Admission Rates (%) | | Adjusted RR (95% CI) | | Adjusted Admission Rates (%) | | Adjusted RR (95% CI) | | | | |
| | 2007–8 | 2015–16 | | | 2007–8 | 2015–16 | | | | | p |
| Overall | 22.5 | 17.3 | 0.77 | (0.68, 0.86) | 14.3 | 11.6 | 0.81 | (0.73, 0.90) | -4.0 | (-11.2, 3.2) | 0.27 |
| Injury | 12.2 | 8.7 | 0.72 | (0.52, 0.91) | 5.8 | 5.3 | 0.93 | (0.74, 1.12) | -21.1 | (-46.4, 4.3) | 0.10 |
| Psychiatric | 24.3 | 21.8 | 0.90 | (0.72, 1.07) | 20.7 | 20.7 | 1.00 | (0.83, 1.17) | -10.2 | (-32.8, 12.4) | 0.38 |
| Upper Respiratory | 17.4 | 11.5 | 0.66 | (0.28, 1.04) | 4.8 | 4.1 | 0.87 | (0.62, 1.12) | -20.8 | (-61.1, 19.5) | 0.31 |
| Abdominal Pain | 27.8 | 18.2 | 0.65 | (0.54, 0.77) | 15.1 | 9.5 | 0.63 | (0.47, 0.78) | 2.8 | (-15.2, 20.7) | 0.76 |
| Leg Symptoms | 18.8 | 11.0 | 0.58 | (0.36, 0.81) | 9.5 | 7.2 | 0.76 | (0.55, 0.96) | -17.6 | (-49.2, 14.0) | 0.27 |
| Chest pain | 45.5 | 29.4 | 0.65 | (0.51, 0.79) | 37.0 | 25.2 | 0.68 | (0.56, 0.80) | -3.6 | (-18.1, 10.9) | 0.63 |
| Neck/Back pain | 10.5 | 12.3 | 1.18 | (0.64, 1.71) | 4.9 | 2.5 | 0.52 | (0.28, 0.76) | 65.3 | (9.5, 121.2) | **0.022** |
| Fever | 23.0 | 18.2 | 0.79 | (0.36, 1.22) | 8.3 | 7.9 | 0.96 | (0.73, 1.20) | -16.9 | (-65.0, 31.2) | 0.49 |
| Nausea/Vomiting/Diarrhea | 25.9 | 20.3 | 0.78 | (0.59, 0.98) | 16.3 | 12.1 | 0.74 | (0.54, 0.93) | 4.5 | (-18.5, 27.5) | 0.70 |
| Shortness of Breath | 45.0 | 51.2 | 1.14 | (0.88, 1.40) | 40.6 | 33.8 | 0.83 | (0.72, 0.94) | 30.4 | (4.9, 55.9) | **0.019** |
| Arm Symptoms | 12.7 | 11.8 | 0.93 | (0.30, 1.55) | 3.9 | 4.9 | 1.24 | (0.74, 1.74) | -31.2 | (-101.4, 39.1) | 0.38 |
| Headache | 9.1 | 8.9 | 0.98 | (0.49, 1.47) | 3.9 | 4.3 | 1.10 | (0.40, 1.80) | -12.3 | (-92.6, 68.0) | 0.76 |
| Skin Complaints | 15.7 | - | - | - | 4.2 | - | - | - | - | - | - |
| Dizziness, syncope | 31.5 | 25.6 | 0.81 | (0.60, 1.03) | 22.5 | 11.7 | 0.52 | (0.37, 0.67) | 29.2 | (3.6, 54.8) | **0.025** |
| Pregnancy Problems | 7.5 | 6.2 | 0.83 | (0.13, 1.53) | 18.3 | 16.8 | 0.92 | (0.46, 1.38) | -8.9 | (-86.9, 69.0) | 0.82 |
| Flank Pain | 13.3 | 8.4 | 0.63 | (0.33, 0.93) | 11.2 | 6.1 | 0.54 | (0.21, 0.87) | 8.5 | (-33.7, 50.6) | 0.70 |
| General Weakness | 52.1 | 54.1 | 1.04 | (0.81, 1.27) | 42.2 | 31.2 | 0.74 | (0.59, 0.89) | 29.9 | (3.5, 56.2) | **0.026** |
| Neurological Symptom | 42.1 | 34.2 | 0.81 | (0.61, 1.02) | 16.0 | 14.7 | 0.92 | (0.47, 1.36) | -10.3 | (-58.6, 38.0) | 0.68 |
| Convulsions | 31.6 | 20.8 | 0.66 | (0.38, 0.93) | 18.1 | 8.1 | 0.45 | (0.23, 0.66) | 21.2 | (-10.6, 53.0) | 0.19 |
| Vaginal Bleeding | - | - | - | - | 10.5 | 7.2 | 0.68 | (0.19, 1.17) | - | - | - |

Adjusted admission rates, adjusted risk ratios, and absolute differences calculated from survey-weighted multivariable logistic regression and marginal estimating methods. All models adjusted for patient and visit characteristics. A negative absolute difference means admission rate reduced by greater proportion among imaged visits compared to visits without advanced imaging. Omitted admission rates (and the corresponding adjusted RR and absolute difference, had <30 observations, which were considered unreliable as recommended by the National Center for Health Statistics.
Abbreviations: RR, risk ratio; CI, confidence interval.

value such as neck/back pain [11] and syncope [12]. An alternative explanation may be that, over time, ED clinicians more liberally use advanced imaging among patients with higher complexity and, therefore, increased likelihood of hospital admission based on information obtained prior to imaging results. In other words, patients who previously would have been hospitalized without ED advanced imaging are now more likely to receive advanced imaging. Unfortunately, our cross-sectional analysis is unable to discern the direction of association. Future studies examining changes in decision making in the clinical context may help further elucidate the underlying drivers of our findings. Nevertheless, our findings do not support the hypothesis that the rise in ED advanced imaging contributed significantly to the decrease in ED hospitalization rates.

Over our study period, advanced imaging use continued to increase. Compared to the 3-fold increase in the decade prior,[2] the increase we observed was much more modest and only significant in 5 of the 20 most common presenting complaints. However, advanced imaging use remained prevalent, particularly for the indications where they may provide limited clinical value, such as neck/back pain [11], syncope [12], and headache [13]. Society guidelines and campaigns such as Choosing Wisely® have sought to reduce advanced imaging use for

these indications. Despite these efforts, we did not observe any downward trend in advanced imaging use. In addition, some limited evidence also suggests that these efforts likely did not contribute to the slowing growth in advanced imaging use [14].

Lastly, though not significantly associated with advanced imaging, ED hospitalization rates have nevertheless declined by 20–30% overall. In complaint-specific analyses, we further found ED hospitalization rates also reduced significantly across visits with and without advanced imaging for primary complaints of abdominal pain, chest pain, and injuries by up to 40%. While our results suggest that increased advanced imaging use may not have contributed to the decline in ED hospitalization rates, other clinical factors, such as outpatient clinical pathways [15, 16], and policy factors, including the rising scrutiny of short-stay admissions and improved access to follow-up as a result of coverage expansion may be have driven the decline in ED hospitalizations [17–19].

Our study has several limitations. National survey data may be susceptible to potential mis-classification of presenting symptoms, ED care received, or discharge diagnoses [20]. How-ever, the misclassification is unlikely to differ across between visits with and without advanced imaging. The cross-sectional nature of the dataset also does not allow us to discern whether visits may be return or repeat ED visits where decisions to pursue advanced imaging would be different from initial ED visits. There were limited data available to account for visits severity and comorbid conditions. Furthermore, we acknowledge that the decision of whether hospi-talize after an ED evaluation is complex may not be fully accounted for in our analysis. Never-theless, the NHAMCS dataset provides unique clinical data not available in typical insurance claims data, such as presenting complaints.

## Conclusion

In this analysis of nationally-representative ED visits, we found that the growth in advanced imaging use has slowed substantially. However, we also found that visits with advanced imag-ing use did not experience a larger reduction in ED hospitalization rates compared to ED visits without advanced imaging. Our finding suggest that the rising advanced imaging use may not have accounted for the decline in ED admission rates, although further research is needed to replicate our findings.

## Supporting information

**S1 Table. Definition of presenting complaints by NCHS reason for visit codes.**
(DOCX)

**S2 Table. Changes in adjusted admission rates comparing 2007–2008 to 2015–2016, only inpatient hospitalization considered admission.**
(DOCX)

**S3 Table. Changes in adjusted admission rates comparing 2007–2008 to 2015–2016, advanced imaging included only CT or MRI.**
(DOCX)

## Author Contributions

**Conceptualization:** Shih-Chuan Chou, Jeremiah D. Schuur.

**Data curation:** Shih-Chuan Chou, Jeremiah D. Schuur.

**Formal analysis:** Shih-Chuan Chou.

**Funding acquisition:** Shih-Chuan Chou.

**Investigation:** Shih-Chuan Chou.

**Methodology:** Shih-Chuan Chou.

**Project administration:** Shih-Chuan Chou, Scott G. Weiner.

**Resources:** Shih-Chuan Chou, Scott G. Weiner.

**Software:** Shih-Chuan Chou.

**Supervision:** Jeremiah D. Schuur, Scott G. Weiner.

**Validation:** Justine M. Nagurney.

**Visualization:** Shih-Chuan Chou, Justine M. Nagurney.

**Writing – original draft:** Shih-Chuan Chou.

**Writing – review & editing:** Shih-Chuan Chou, Justine M. Nagurney, Jeremiah D. Schuur, Scott G. Weiner.

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
