## [Decision Letter · Decision Letter 0]

6 Jul 2020

PONE-D-20-04321

Advanced Imaging and Trends in Hospitalizations from the Emergency Department

PLOS ONE

Dear Dr. Chou,

Thank you for submitting your manuscript to PLOS ONE. After careful consideration, we feel that it has merit but does not fully meet PLOS ONE’s publication criteria as it currently stands. Therefore, we invite you to submit a revised version of the manuscript that addresses the points raised during the review process.

We look forward to receiving your revised manuscript.

Kind regards,

Steve Lin

Academic Editor

PLOS ONE

Journal Requirements:

2. Please refrain from stating p values as 0.00, either report the exact value or employ the format p<0.001.

3. Please remove your figures from within your manuscript file, leaving only the individual TIFF/EPS image files, uploaded separately.  These will be automatically included in the reviewers’ PDF.

Reviewers' comments:

Reviewer's Responses to Questions

**Comments to the Author**

1. Is the manuscript technically sound, and do the data support the conclusions?

Reviewer #1: Yes

2. Has the statistical analysis been performed appropriately and rigorously? 

Reviewer #1: Yes

3. Have the authors made all data underlying the findings in their manuscript fully available?

Reviewer #1: Yes

4. Is the manuscript presented in an intelligible fashion and written in standard English?

Reviewer #1: Yes

5. Review Comments to the Author

Reviewer #1: This manuscript is very well-written and addresses an important topic regarding healthcare resource utilization. Having said that, this issue is complex and I have a few suggestions that would potentially help the paper reflect this complexity.

1. Hypothesis

I think your hypothesis - that advanced ED imaging would lead to a decline in admission rates - needs a bit more rationale earlier in the paper. Intuitively, and as you later mention in the discussion, advanced imaging could instead lead to an increase in hospitalization if more subtle but possibly dangerous pathology was identified. Although your paper did not show any statistical difference for imaging vs. hospitalization in either direction, I think some justification for your hypothesis would help the reader follow the paper.

2. Confounding variables

There are several variables that affect the decision to admit a patient to hospital. Of course, the disposition of patients at either end of the spectrum are obvious (ie. a fractured toe vs. a patient in septic shock), but much of emergency medicine operates in a grey area where two different providers could make two different, and equally justifiable, decisions. This is difficult to capture in a binary "admitted vs. discharge" outcome measure.

Additionally, admission can sometimes be a product of patient preference or lack of outpatient supports. I think that you have done a good job explaining some of these variables in the conclusion ("outpatient clinical pathways, and policy factors...") but can better underscore the many factors which are at play in the decision to admit a patient to hospital, and include this is a major limitation to your study.

3. Methods

How were patients who were seen in the ED and then discharged with a plan/appointment to return the next day for advanced imaging dealt with? Was this information reflected in the database? This is a relatively common practice in the EDs I have worked at for patients seen in the evening/overnight without critically emergent differential diagnoses.

4. Results

No further suggestions here - I was happy to see you include relevant high-level results for subgroups (ie. excluding observation admissions and excluding ultrasound) - I think this adds credence to the arguments brought forth in the paper.

5. Conclusion

You suggest that advanced imaging may not play a substantial role in the decline of admission rates. Despite your well-executed study, I am still not convinced of this, largely because of the many confounding variables, and so I think it would be more appropriate to state that more research is needed.

6. PLOS authors have the option to publish the peer review history of their article (what does this mean?). If published, this will include your full peer review and any attached files.

Reviewer #1: **Yes: **Shaun Mehta

---

## [Author Response · Author response to Decision Letter 0]

5 Aug 2020

Reviewer #1: This manuscript is very well-written and addresses an important topic regarding healthcare resource utilization. Having said that, this issue is complex and I have a few suggestions that would potentially help the paper reflect this complexity.

1. Hypothesis

I think your hypothesis - that advanced ED imaging would lead to a decline in admission rates - needs a bit more rationale earlier in the paper. Intuitively, and as you later mention in the discussion, advanced imaging could instead lead to an increase in hospitalization if more subtle but possibly dangerous pathology was identified. Although your paper did not show any statistical difference for imaging vs. hospitalization in either direction, I think some justification for your hypothesis would help the reader follow the paper.

 Thank you for this suggestion, we reorganized the introduction to refocus the study’s emphasis on examining the value of increased advanced imaging through reduction in 

 hospital admissions. 

2. Confounding variables

There are several variables that affect the decision to admit a patient to hospital. Of course, the disposition of patients at either end of the spectrum are obvious (ie. a fractured toe vs. a patient in septic shock), but much of emergency medicine operates in a grey area where two different providers could make two different, and equally justifiable, decisions. This is difficult to capture in a binary "admitted vs. discharge" outcome measure.

Additionally, admission can sometimes be a product of patient preference or lack of outpatient supports. I think that you have done a good job explaining some of these variables in the conclusion ("outpatient clinical pathways, and policy factors...") but can better underscore the many factors which are at play in the decision to admit a patient to hospital, and include this is a major limitation to your study.

 In this revision, we acknowledge the limitation of our data to account for the complex decision of admitting a patient more explicitly. 

 “Furthermore, we acknowledge that the decision of whether hospitalize after an ED evaluation is complex may not be fully accounted for in our analysis.”

3. Methods

How were patients who were seen in the ED and then discharged with a plan/appointment to return the next day for advanced imaging dealt with? Was this information reflected in the database? This is a relatively common practice in the EDs I have worked at for patients seen in the evening/overnight without critically emergent differential diagnoses.

 We are not aware that planned return ED visits is a commonplace practice, at least in the community sites where we work. Nevertheless, the NHAMCS data is consisted of a 

 cross-sectional sample of ED visits from EDs that were surveyed annually. The patients are thus not followed longitudinally across visits. Therefore, NHAMCS is unable to identify 

 visits that may be “return” or “repeat” visits. We additionally noted this limitation as considerations for imaging use would certainly be different for repeat or return visits. 

 “The cross-sectional nature of the dataset also does not allow us to discern whether visits may be return or repeat ED visits where decisions to pursue advanced imaging would be 

 different from initial ED visits.”

4. Results

No further suggestions here - I was happy to see you include relevant high-level results for subgroups (ie. excluding observation admissions and excluding ultrasound) - I think this adds credence to the arguments brought forth in the paper.

Thank you.

5. Conclusion

You suggest that advanced imaging may not play a substantial role in the decline of admission rates. Despite your well-executed study, I am still not convinced of this, largely because of the many confounding variables, and so I think it would be more appropriate to state that more research is needed.

 We adjusted our tone in the CONCLUSION section to be more align our finding as preliminary and needs further replication. However, we do believe that our study design is 

 stronger exploratory compared to studies that simply observed that the decline in admission rates and the increase in advanced imaging occurred concurrently.

---

## [Decision Letter · Decision Letter 1]

31 Aug 2020

Advanced Imaging and Trends in Hospitalizations from the Emergency Department

PONE-D-20-04321R1

Dear Dr. Shih-Chuan Chou,

We’re pleased to inform you that your manuscript has been judged scientifically suitable for publication and will be formally accepted for publication once it meets all outstanding technical requirements.

Kind regards,

Steve Lin

Academic Editor

PLOS ONE

Additional Editor Comments (optional):

Please consider the last few suggestions made by the reviewers.

Reviewers' comments:

Reviewer's Responses to Questions

**Comments to the Author**

1. If the authors have adequately addressed your comments raised in a previous round of review and you feel that this manuscript is now acceptable for publication, you may indicate that here to bypass the “Comments to the Author” section, enter your conflict of interest statement in the “Confidential to Editor” section, and submit your "Accept" recommendation.

Reviewer #1: All comments have been addressed

Reviewer #2: All comments have been addressed

2. Is the manuscript technically sound, and do the data support the conclusions?

Reviewer #1: Yes

Reviewer #2: Yes

3. Has the statistical analysis been performed appropriately and rigorously? 

Reviewer #1: Yes

Reviewer #2: Yes

4. Have the authors made all data underlying the findings in their manuscript fully available?

Reviewer #1: Yes

Reviewer #2: Yes

5. Is the manuscript presented in an intelligible fashion and written in standard English?

Reviewer #1: Yes

Reviewer #2: Yes

6. Review Comments to the Author

Reviewer #1: Thank you for taking the time to address my comments. I feel that you have addressed everything comprehensively and that this paper adds value to the body of scientific literature as it relates to emergency medicine.

Reviewer #2: This is a well designed study on a relevant topic. This observational study has a strong analytical methods to answer the question of whether increases in imaging use may have resulted in decreased hospital admissions. The comments of the reviewers were addressed.

A few outstanding comments:

1. There is an issue with the wording here: "we acknowledge that the decision of whether hospitalize after an ED evaluation is complex may not be fully accounted for in our"

2. The conclusion does not need to suggest a specific type of study such as a "replication study". "Replicate" could be changed for "support".

7. PLOS authors have the option to publish the peer review history of their article (what does this mean?). If published, this will include your full peer review and any attached files.

Reviewer #1: **Yes: **Shaun Mehta

Reviewer #2: **Yes: **Samuel Vaillancourt

---

## [Editor Report · Acceptance letter]

4 Sep 2020

PONE-D-20-04321R1 

Advanced Imaging and Trends in Hospitalizations from the Emergency Department 

Dear Dr. Chou:

I'm pleased to inform you that your manuscript has been deemed suitable for publication in PLOS ONE. Congratulations! Your manuscript is now with our production department. 

Kind regards, 

on behalf of

Dr. Steve Lin 

Academic Editor

PLOS ONE